# Impact of *Zinnia elegans* Cultivation on the Control Efficacy and Distribution of *Aphidius colemani* Viereck (Hymenoptera: Braconidae) against *Aphis gossypii* Glover (Hemiptera: Aphididae) in Cucumber Greenhouses

**DOI:** 10.3390/insects15100807

**Published:** 2024-10-15

**Authors:** Eun-Jung Han, Sung-Hoon Baek, Jong-Ho Park

**Affiliations:** 1International Technology Cooperation Center, Rural Development Administration, Jeonju 54875, Republic of Korea; hejs2@korea.kr; 2Entomology Program, Department of Agricultural Biotechnology, Seoul National University, Seoul 00826, Republic of Korea; 3Research & Development Center, EPINET, Anyang 14057, Republic of Korea; shbaek007@hotmail.com; 4Organic Agricultural Division, National Institute of Agricultural Sciences, Wanju 55635, Republic of Korea

**Keywords:** cotton aphid, colemani wasp, companion plant, conservation biological control, habitat management

## Abstract

This study investigated the potential for enhancing the efficacy of the biological control agent, *Aphidius colemani*, against the cotton aphid, *Aphis gossypii*, by planting *Zinnia elegans* flowers alongside cucumbers in greenhouses. Three treatments were examined: (1) the combined use of *Z. elegans* and *A. colemani*; (2) *A. colemani* alone; and (3) a control without either. Our results indicated that the *Z. elegans* and *A. colemani* combination maintained a lower *A. gossypii* population compared to other treatments. While *A. colemani* alone initially suppressed the *A. gossypii* population, it soon increased. This study concluded that integrating *Z. elegans* with the biological control agent could more effectively manage aphid populations in cucumber greenhouses.

## 1. Introduction

The cotton aphid *Aphis gossypii* Glover is a pest of global distribution that infests a variety of crops, leading to economic damage. It has been documented to attack over 320 host plants across 46 families [1], with a particular preference for plants in the Solanaceae family [2]. Continuous feeding by *A. gossypii* results in shoot deformities, stunted growth, reduced fruit set, and delayed fruit maturation in cucumber plants. Additionally, *A. gossypii* acts as a vector for viruses from the Potyviridae family, including the cucumber mosaic virus, watermelon mosaic virus, papaya ring spot virus, and so on [3].

Once established in agricultural fields, *A. gossypii* populations rapidly increase due to their high reproductive rate and short generation time [4], necessitating effective control measures. However, the excessive use of chemical insecticides has led to *A. gossypii* developing resistance against compounds such as pirimicarb, endosulfan, pyrethroid, and imidacloprid [5,6,7], creating the need for environmentally friendly alternatives.

*Aphidius colemani* is recognized as a natural enemy that parasitizes various aphid species, including *A. gossypii* [8]. It effectively suppresses these aphid populations [9] with remarkable search capabilities [10]. The release of *A. colemani* has successfully managed *A. gossypii* populations in commercial cucumber greenhouses [11,12]. After its success in these experiments, diverse studies were conducted regarding its release’s timing and number, the toxicity assessment of its chemical or organic pesticides, and the compatibility testing of these materials [11,12,13,14,15,16,17,18,19,20]. However, there are still limited studies on the establishment and conservation of *A. colemani* populations after the species’ introduction in agricultural fields.

Over the past two decades, research on conservation biological control (CBC) has aimed to conserve natural enemies by providing the resources and conditions that they need to establish in agricultural environments and enhance pest control [21,22,23,24]. Charles-Tollerup (2013) categorized CBC implementation into five categories [25]: (1) providing supplemental food resources like pollen, floral or extra-floral nectar, and alternative prey; (2) minimizing pesticide application; (3) managing secondary enemies; (4) optimizing host plant attributes; and (5) mitigating negative cultural practices or tactics.

Agricultural fields practicing monoculture cultivation are typically seen as having a relative scarcity of sugar sources, such as floral and extra-floral nectar [26]. For parasitoid insects, alternative food sources could potentially enhance their longevity [26,27,28], oviposition [29,30], and parasitic rates [31,32]. Despite numerous studies demonstrating the impact of alternative food sources, like sugars, on the life history traits and parasitism improvement of parasitoid natural enemies, the widespread application of techniques for providing alternative foods in greenhouse environments remains limited. Benson and Labbe (2021) identified one reason for this limitation as the lack of assessment regarding the pest control efficacy of the natural enemies that utilize the alternative foods under greenhouse conditions [33]. Therefore, this study aimed to evaluate the synergistic effect of companion planting with *Z. elegans* as an alternative food source in a cucumber greenhouse with *A. colemani*. The hypothesis proposed that incorporating *Z. elegans* could enhance pest control efficacy by providing adult parasitoids with alternative food through its flowers. To test this hypothesis, this study assessed the influence of *Z. elegans* on the spatial distribution of *A. gossypii* and its mummies parasitized by *A. colemani* within a greenhouse.

## 2. Materials and Methods

### 2.1. Cucumber Cultivation Methods

Experimental studies were conducted from 2016 to 2017 to assess the impact of *Z. elegans* cultivation on the suppression of *A. gossypii* density by *A. colemani*. The research was carried out in plastic greenhouses at the organic research farm of the National Institute of Agricultural Science in Iseo-myeon, Wanju-gun, Jeollabuk-do, the Republic of Korea (N 35.49.35.85, E 127.02.52.64). Each greenhouse unit, covering an area of 260 m^2^, was internally partitioned into four rows, each being 0.8 m wide, with a 0.4 m gap between them. Mesh screens were fitted on both sides of the greenhouse and at the front and rear entrances.

The experimental plant was the cucumber *Cucumis sativus*, specifically the cultivar “Cheongeoram Baekdadaki”. Cucumber seeds were planted in 72-cell plastic trays and grown in a glasshouse until they developed 2–3 leaves. In the spring cultivation trials, cucumber seedlings were transplanted on 8 April 2016 and 5 April 2017. For the fall cultivation trials, transplantation took place on 30 August 2016 and 1 September 2017.

### 2.2. Changes in the Density of A. gossypii and A. colemani

The effects of *A. colemani* and *Z. elegans* were assessed in the plastic greenhouses. Experimental treatments included (1) the simultaneous application of *A. colemani* and cultivation of *Z. elegans* (referred to as “Parasitoid-Zinnia”), (2) the application of *A. colemani* alone (referred to as “Parasitoid”), and (3) a control (i.e., no *A. colemani* and *Z. elegans*).

*A. colemani* were released after being reared on banker plants, which consisted of barley plants inoculated with *Aphis glycine* Matsumura. In the spring trials, these banker plants were introduced into the greenhouse two weeks after the transplantation of cucumber seedlings. In the fall trials, the introduction of banker plants coincided with the transplantation of cucumber seedlings. Each greenhouse unit received three banker plants, with moisture supplied twice daily through drip hoses to prevent wilting. *Z. elegans* was sown on both sides of the greenhouse six weeks prior to cucumber transplantation and cultivated alongside cucumber throughout the cucumber cultivation period.

The density of *A. gossypii* and mummies parasitized by *A. colemani* was observed and recorded. In a spring trial in 2016, between 104 and 122 cucumber plants were selected and marked at regular intervals (4 m by 4 m). In subsequent trials, 20 cucumber plants per treatment were randomly selected. From the fully expanded first leaf derived from the shoot tip of each selected plant, *A. gossypii* and parasitized mummies were counted every four leaves through visual inspection, and the average value within the selected plant was used for analyses. The parasitism rate was calculated by dividing the mean number of mummies by the sum of the mean number of *A. gossypii* and mummies. The random assignment of greenhouses across four trials was employed to minimize potential confounding effects between the greenhouse and treatment.

To analyze changes in the density of *A. gossypii*, parasitized mummies, and parasitism rates over time, the repeated measures ANOVA (analysis of variance) test was applied using ‘car’ and ‘emmeans’ packages of R version 4.4.1. A linear model was fitted for each responsible variable, incorporating the main effects of treatment and observation times (days after cucumber transplanted, DACT), as well as their interaction using the Type Ⅲ sum of squares. To evaluate differences among treatments at each observation time, estimated marginal means were calculated. Comparisons among treatments were performed within each DACT level, and the Bonferroni correction was applied to *p*-values to control for multiple comparisons. The overall significance level was set at α = 0.05. The parasitism rate was transformed into an arcsine value before conducting ANOVA due to a violation of the normality assumption.

### 2.3. Analyzing Spatial Distribution Patterns of A. gossypii and A. colemani

The average number of *A. gossypii* and its mummies per leaf from the spring trial in 2016 and 2017 were used for the analysis of spatial distribution. Spatial analysis by distance indices (SADIE) was employed, using the index of aggregation (*I*_a_), to assess the spatial distributions of aphids, parasitized mummies, and parasitism rates [34].
*I*_a_ = *D*/*E*_a_

In this equation, *D* represents the minimum total distance that individuals should move to achieve a uniformly distributed pattern within each sample unit, and *E*_a_ denotes the mean value for *D* under randomization. The value of *I*_a_ = 1 signifies a randomly arranged distribution, while *I*_a_ > 1 indicates an aggregated distribution and *I*_a_ < 1 indicates a uniform distribution. The associated probability (*P*_a_) of this aggregation index was calculated using randomization tests at *p =* 0.05. The significance test for spatial patterns was executed with formal randomization tests with SADIEShell version 1.22 (Rothamsted Experimental Station; Harpenden, Herts, UK) [34]. The distribution patterns of *A. gossypii*, *A. colemani*, and their parasitism rates were visualized using ArcGIS version 10.1 (ESRI; Redlands, CA, USA).

## 3. Results

### 3.1. Population Dynamics of A. gossypii and A. colemani in Spring Cucumbers

A repeated-measures ANOVA revealed that the main effect of DACT was highly significant (*F* = 219.78, df = 3, 1347, *p* < 0.0001), as was the interaction between treatment and DACT (*F* = 53.97, df = 6, 1344, *p* < 0.0001), indicating significant temporal variation and treatment effects over time. Pairwise contrasts at different DACT levels showed no significant differences among treatments at 20 DACT and 31 DACT, but significant differences emerged at 42 DACT and 59 DACT (Figure 1a).

The parasitoid treatment maintained a consistently low occurrence of *A. gossypii* by 42 DACT and subsequently experienced a rapid increase to 156 individuals per leaf by 59 DACT. Conversely, in the parasitoid–zinnia plot, *A. gossypii* consistently maintained a low density of seven individuals per leaf throughout the survey period, following their initial observation on 20 DACT (Figure 1a).

The occurrence of *A. gossypii* in 2017 showed a higher density compared to 2016; however, the progression of density exhibited a similar pattern (Figure 1b). The repeated-measures ANOVA analysis of aphid density in the spring of 2017 showed that DACT (*F* = 102.597, df = 4, 296, *p* < 0.0001) and the interaction between treatment and DACT (*F* = 10.16, df = 8, 292, *p* < 0.0001) had statistically significant effects. Bonferroni corrected comparisons indicated that there were no significant differences between the three treatments until 50 DACT. However, starting from 56 DACT, differences between treatments became apparent (Figure 1b).

In the control plot, the density of *A. gossypii* began to increase at 43 DACT, reaching 395 individuals per leaf by 56 DACT, followed by a decrease due to crop damage caused by the high density of *A. gossypii*. In the parasitoid plot, *A. gossypii* remained low until 43 DACT, subsequently increasing to 493 individuals per leaf by 63 DACT. In the parasitoid–zinnia plot, *A. gossypii* gradually increased after its initial occurrence at 36 DACT, peaking at 120 individuals per leaf by 56 DACT before declining again.

The density of mummies attacked by parasitoids remained consistently low across all plots in the 2016 spring trial (Figure 2a). The number of mummies significantly affected by DACT (*F* = 194.8, df = 3, 1347, *p* < 0.0001) and the interaction between treatment and DACT (*F* = 47.22, df = 6, 1344, *p* < 0.0001) were statistically significant. There were no significant differences between treatments at 20, 31, and 42 DACT but significant differences emerged at 59 DACT across all treatment contrasts (Figure 2a).

The parasitism rate showed significant changes over time (DACT) and varied by treatment, with notable differences between groups emerging at specific time points, particularly at 42 DACT and 59 DACT.

In 2017, higher densities of mummies were observed compared to the previous year (Figure 2b). This increase in mummy density was associated with the significant effect of DACT (*F* = 38.67, df = 4, 296, *p* < 0.0001) and its interaction with treatment (*F* = 10.15, df = 8, 292, *p* < 0.0001), suggesting that the impacts of treatments varied over time. Pairwise comparisons revealed significant differences between treatment groups at later time points (56 and 63 DACT), indicating that the temporal changes in parasitoid populations were strongly influenced by the applied treatments. When comparing parasitism rates, significant differences were observed after 56 and 63 DACT. Excluding the period of rapid increase in the parasitism rate at 63 DACT, the parasitism rate was significantly higher in the control plot (Figure 3b).

### 3.2. Population Dynamics of A. gossypii and A. colemani in Fall Cucumber

The densities of *A. gossypii* in the cucumber plastic greenhouse during the fall trial in 2016 showed significant variation influenced by DACT (*F* = 75.69, df = 2, 178, *p* < 0.0001) and the interaction between treatment and DACT (*F* = 23.92, df = 4, 176, *p* < 0.0001, Figure 4a).

Unlike the spring trials, the fall trials showed a notable increase in *A. gossypii* density within a shorter period. Specifically, in the parasitoid and control plots, *A. gossypii* densities exhibited a substantial rise from 24 DACT to 35 DACT, reaching considerable populations of 2228 and 2573 individuals per leaf, respectively. In the parasitoid–zinnia plot, *A. gossypii* were not observed until 24 DACT, but it increased to 225 individuals per leaf by 35 DACT.

In the fall trial of 2017, *A. gossypii* exhibited significant variation influenced by DACT (*F* = 62.94, df = 6, 234, *p* < 0.0001) and the interaction between treatment and DACT (*F* = 9.08, df = 6, 234, *p* < 0.0001). Consistent with the findings from 2016, higher densities of *A. gossypii* were observed in the control plot and the parasitoid plot, while the density remained consistent in the parasitoid–zinnia plot, staying below 60 (Figure 3b).

In the fall trial of 2016, the analysis of parasitized mummy density showed that DACT had a significant effect (*F* = 12.32, df = 2, 178, *p* < 0.0001); however, the treatment and its interaction with DACT were not significant. For the parasitism rate, the interaction between treatment and DACT was significant (*F* = 9.08, df = 4, 176, *p* < 0.0001), with significant differences being observed at 35 DACT (Figure 5a and Figure 6a).

In the fall of 2017, parasitized mummies exhibited significant variation influenced by DACT (*F* = 60.17, df = 3, 237, *p* < 0.0001) and its interaction with treatment (*F* = 16.48, df = 6, 234, *p* < 0.0001). Parasitized mummies were not observed in any treatment plots during the initial observation at 10 and 27 DACT. All plots showed low densities until 27 DACT. The parasitoid plot had significantly higher mummy densities, reaching up to 121.8 individuals per leaf, while the parasitoid–zinnia plot had a density of 20.8 individuals per leaf, and the control plot had only 4.6 individuals per leaf by 40 DACT. The parasitism rate of aphids also showed significant variation influenced by DACT (*F* = 18.87, df = 3, 237, *p* < 0.0001) and its interaction with treatment (*F* = 7.97, df = 6, 234, *p* < 0.0001), exhibiting significant differences between treatments after 40 DACT (Figure 5b and Figure 6b).

### 3.3. Spatial Distribution of A. gossypii and A. colemani

The spatial distributions of *A. gossypii* and its mummy densities parasitized by the aphid parasitoid, as well as the parasitism rate, were analyzed using SADIE among the plots in the spring trial of 2016. The spatial distribution of *A. gossypii* exhibited a random distribution at the onset of occurrence across all treatment plots, transitioning to an aggregated distribution as the density increased, subsequently reverting to a random distribution as dispersion occurred.

At 31 DACT, *A. gossypii* exhibited an aggregated distribution in the control plot (*I*_a_ = 2.501, *P*_a_ = 0.0128). By 42 DACT, this species showed an aggregated distribution across all treatment plots: the parasitoid–zinnia plot (*I*_a_ = 2.197, *P*_a_ = 0.0128), the parasitoid plot (*I*_a_ = 2.297, *P*_a_ = 0.0128), and the control plot (*I*_a_ = 1.897, *P*_a_ = 0.0128) (Table 1; Figure 7).

Parasitized mummies were observed to be aggregated in the control plot at both 31 DACT (*I*_a_ = 2.039, *P*_a_ = 0.0128) and 42 DACT (*I*_a_ =0.588, *P*_a_ = 0.0128), although they predominantly exhibited a random distribution (Figure 7).

The spatial distribution of the parasitism rates of *A. gossypii* by parasitoids displayed a pattern distinct from the distribution of *A. gossypii* and mummies. In the parasitoid–zinnia plot, the parasitism rate was uniformly distributed during the observation at 59 DACT (*I*_a_ = 0.519, *P*_a_ = 0.9872). In contrast, in the parasitoid plot, the parasitism rate remained randomly distributed throughout the observation period. However, in the control plot, it demonstrated an aggregated distribution on both 42 DACT and 59 DACT.

The spatial distribution and aggregation of *A. gossypii*’s densities, its mummy densities, and its parasitism rates showed similar patterns to the ones in 2016 (Appendix A).

## 4. Discussion

Our research results indicate the importance of incorporating non-crop plants, such as *Z. elegans* flower strips, for effective biological control using parasitoid natural enemies in greenhouse environments. In the spring trial of 2016, the parasitoid–zinnia plot demonstrated that the density of *A. gossypii* consistently remained below the economic injury threshold level of 10 aphids per leaf [35] throughout the observation period. This trend was also observed in other three trials where the density of *A. gossypii* in the parasitoid–zinnia plot was significantly lower compared to the other treatment plots.

However, when *A. colemani* were solely introduced with banker plants in the greenhouse, the effectiveness of its control against *A. gossypii* decreased rapidly over time after roughly seven weeks from its release. Kim and Kim [16] also reported that the pest control efficacy of *A. colemani* using a banker plant system lasted for only for 49 days. Similarly, Karacaoglu et al. [36] highlighted the need for multiple releases of *A. colemani* to effectively control peach aphid populations on eggplants. Successful aphid management and the necessity for multiple releases of *A. colemani* in greenhouse conditions have also been reported in studies involving melons [37] and cucumbers [13].

While augmentative biological controls involving *A. colemani* enhance pest management, they also escalate control costs. Thus, farmers generally hesitate to implement this strategy in their greenhouses due to the high costs and restrictions on the use of inexpensive chemical pesticides. To solve this issue, manipulative biological control strategies have been developed. One such strategy involves providing an alternative food source for the target biological control agent. Tylianakis et al. [38] reported the increased lifespan and fecundity of two parasitic wasps, *Aphidius rhopalosiphi* De Stefani Perez and *Diaeretiella rapae* (McIntosh), when provided with buckwheat pollen, sugar, and flowers of white sage and lantana. Adult parasitic wasps from the Braconidae family are known to require food resources beyond their hosts. The consumption of plant-derived nectar, pollen, and honeydew from aphid colonies has been reported to influence factors such as lifespan, fecundity, parasitism rate, and egg maturation [31,38]. This study further demonstrated that *Z. elegans* can help to establish and maintain populations of released *A. colemani* and manage its pest, *A. gossypii*, under greenhouse conditions.

In this study, the spatial distribution of *A. gossypii* showed an aggregative tendency corresponding to its increasing density across all treatment plots. This finding is consistent with previous research conducted on various crops, including watermelon [39], chili [40], and cotton [41]. This aggregated pattern in spatial distributions is common in insects, including natural enemies, due to food preferences, environmental adaptation, and responses to limited resources [42]. In the control condition of this study (i.e., no release of *A. colemani*), the distribution of mummies also showed aggregated distributions, indicating the likelihood of encounters and predation between natural enemies and their prey [43]. However, the spatial distributions of *A. gossypii* mummies changed to random, indicating the presence of interactions between individuals of *A. colemani*, a proposition previously reported in *Aphidius ervi* Haliday [44]. In conditions providing alternative food, *Z. elegans*, the parasitism rate of *A. gossypii* by *A. colemani* showed a uniform distribution after 59 days from the release of *A. colemani*, indicating the successful control of *A. gossypii* and establishment of *A. colemani*.

This study highlights the potential of companion planting with *Z. elegans*, in conjunction with the application of *A. colemani*, to enhance pest control efficiency in cucumber greenhouses. However, a comprehensive assessment of the impact of *Z. elegans* on both *A. gossypii* and *A. colemani* populations was not fully explored in this study, as the evaluation of pest control efficacy against *A. gossypii* was conducted only within 59 days of cucumber transplanting. Beyond this period, the cucumbers became heavily contaminated by *A. gossypii* and could not survive further under the control conditions of this study. For the practical implementation of the findings from this study, additional research spanning the entire cucumber cultivation period would be necessary in commercial greenhouses.

## Figures and Tables

**Figure 1 insects-15-00807-f001:**
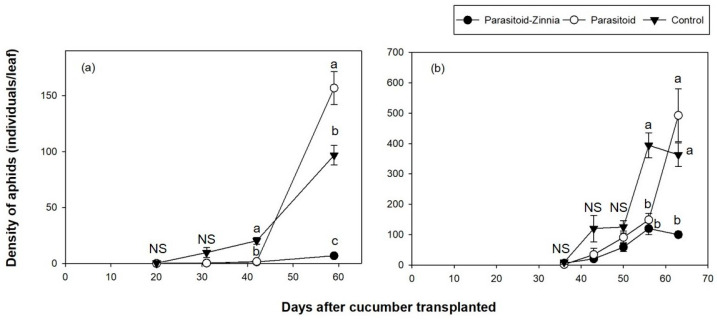
The densities (mean ± SE) of *A. gossypii* in cucumber greenhouses during the spring seasons of 2016 (**a**) and 2017 (**b**) according to different treatments (i.e., the simultaneous application of *A. colemani* and cultivation of *Z. elegans* (parasitoid–zinnia), the application of *A. colemani* alone (parasitoid), and the control (no *A. colemani* and *Z. elegans*)). The values on the same date followed by different letters are significantly different at the 95% confidential level. “NS” means that values are not significantly different at the 95% confidential level.

**Figure 2 insects-15-00807-f002:**
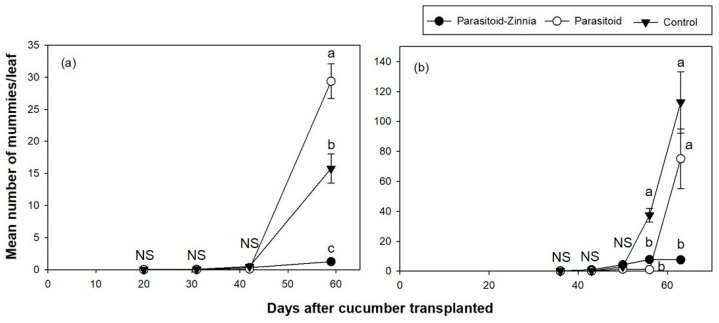
Th densities (mean ± SE) of parasitized mummies per leaf in cucumber greenhouses during the spring seasons of 2016 (**a**) and 2017 (**b**) according to different treatments (i.e., the simultaneous application of *A. colemani* and cultivation of *Z. elegans* (parasitoid–zinnia), the application of *A. colemani* alone (parasitoid), and the control (no *A. colemani* and *Z. elegans*)). The values on the same date followed by different letters are significantly different at the 95% confidential level. “NS” means that values are not significantly different at the 95% confidential level.

**Figure 3 insects-15-00807-f003:**
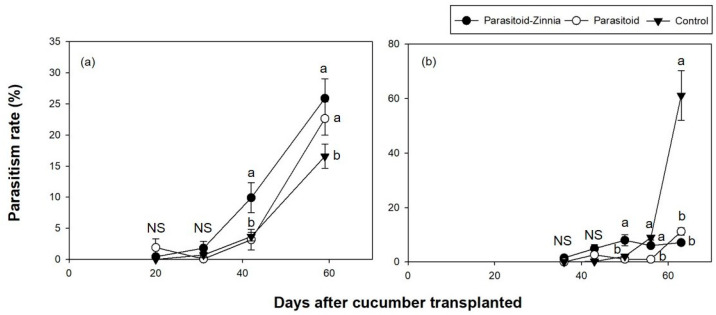
A comparison of *A. gossypii* parasitism rates (mean ± SE) on cucumber in a greenhouse during the spring seasons of 2016 (**a**) and 2017 (**b**) under three different treatments (i.e., the simultaneous application of *A. colemani* and cultivation of *Z. elegans* (parasitoid–zinnia), the application of *A. colemani* alone (Parasitoid), and the control (no *A. colemani* and *Z. elegans*)). The values on the same date followed by different letters are significantly different at the 95% confidential level. “NS” means that values are not significantly different at the 95% confidential level.

**Figure 4 insects-15-00807-f004:**
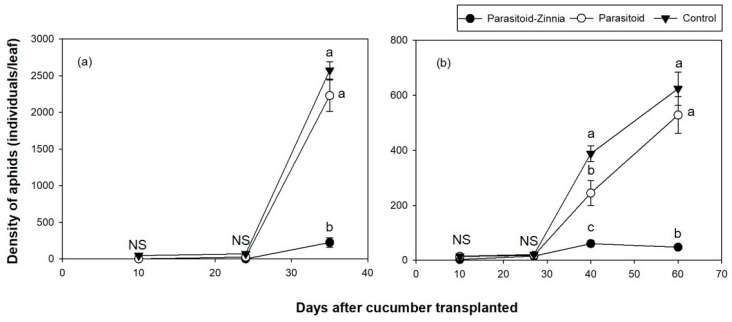
The densities (mean ± SE) of *A. gossypii* in cucumber greenhouses during the fall seasons of 2016 (**a**) and 2017 (**b**) according to different treatments (i.e., the simultaneous application of *A. colemani* and cultivation of *Z. elegans* (parasitoid–zinnia), the application of *A. colemani* alone (parasitoid), and the control (no *A. colemani* and *Z. elegans*)). The values on the same date followed by different letters are significantly different at the 95% confidential level. “NS” means that values are not significantly different at the 95% confidential level.

**Figure 5 insects-15-00807-f005:**
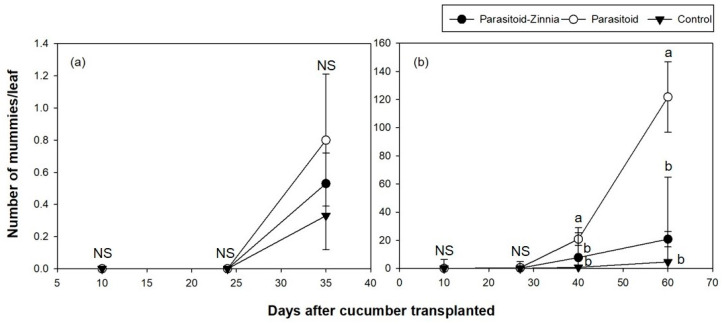
The densities (mean ± SE) of parasitized mummies per leaf in cucumber greenhouses during the fall seasons of 2016 (**a**) and 2017 (**b**) according to different treatments (i.e., the simultaneous application of *A. colemani* and cultivation of *Z. elegans* (parasitoid–zinnia), the application of *A. colemani* alone (parasitoid), and the control (no *A. colemani* and *Z. elegans*)). The values on the same date followed by different letters are significantly different at the 95% confidential level. “NS” means that values are not significantly different at the 95% confidential level.

**Figure 6 insects-15-00807-f006:**
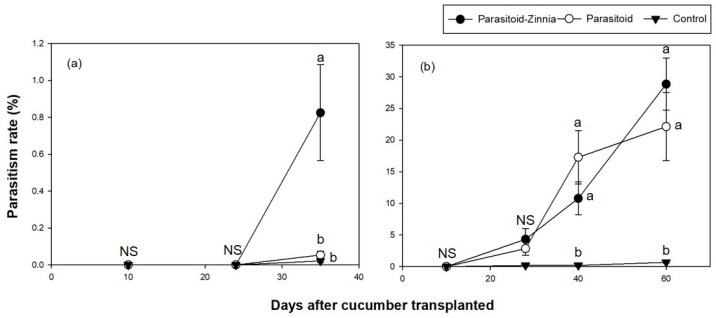
A comparison of *A. gossypii* parasitism rates (mean ± SE) on cucumber in a greenhouse during the fall seasons of 2016 (**a**) and 2017 (**b**) under three different treatments (i.e., the simultaneous application of *A. colemani* and cultivation of *Z. elegans* (parasitoid–zinnia), the application of *A. colemani* alone (Parasitoid), and the control (no *A. colemani* and *Z. elegans*)). The values on the same date followed by different letters are significantly different at the 95% confidential level. “NS” means that values are not significantly different at the 95% confidential level.

**Figure 7 insects-15-00807-f007:**
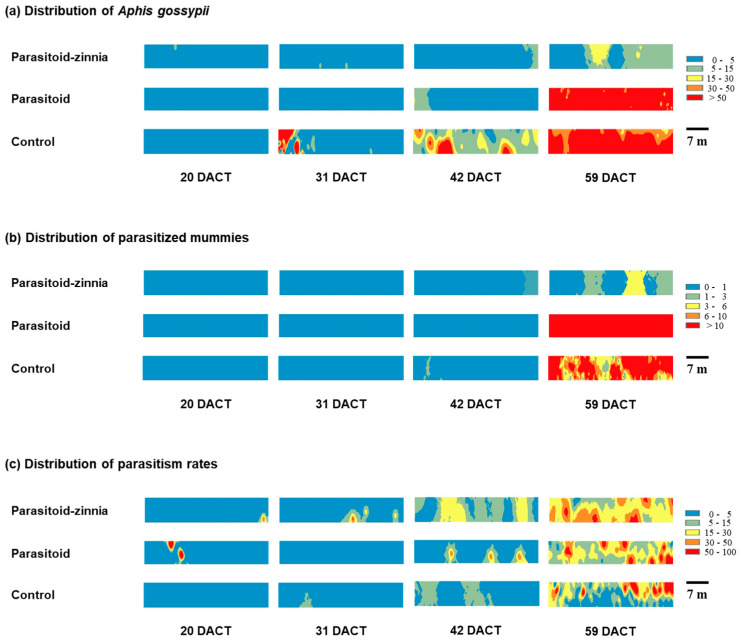
The distribution maps of *A. gossypii* densities (**a**), its parasitized mummy densities (**b**), and parasitism rates (**c**) under different treatments (i.e., the simultaneous application of *A. colemani* and cultivation of *Z. elegans* (parasitoid–zinnia), the application of *A. colemani* alone (parasitoid), and the control (no *A. colemani* and *Z. elegans*)) in the spring of 2016. DACT indicates days after cucumber transplantation.

**Table 1 insects-15-00807-t001:** The spatial distribution patterns and their related parameters of *A. gossypii*, as well as their parasitized mummies and parasitism rates, using SADIE in three treatments during the spring trials of 2016.

Date (DACT)	Treatment								
	Natural Enemies-Zinnia			Natural Enemies			Control		
	*I* _a_	*P* _a_	Pattern	*I* _a_	*P* _a_	Pattern	*I* _a_	*P* _a_	Pattern
*Aphis gossypii*									
20	1.07	0.37	Random	N/A *	N/A	-	1.25	0.18	Random
31	1.05	0.35	Random	1.64	0.09	Random	2.50	0.01	Aggregated
42	2.19	0.01	Aggregated	2.30	0.01	Aggregated	1.90	0.01	Aggregated
59	1.22	0.18	Random	1.00	0.40	Random	1.09	0.33	Random
Parasitized mummy									
20	0.90	0.55	Random	1.30	0.17	Random	N/A	N/A	-
31	1.29	0.18	Random	N/A	N/A	-	2.04	0.01	Aggregated
42	1.48	0.09	Random	0.63	0.90	Random	0.59	0.01	Aggregated
59	1.08	0.35	Random	1.13	0.27	Random	1.76	0.64	Random
Parasitism rate									
20	N/A	N/A	-	1.30	0.17	Random	N/A	N/A	-
31	1.03	0.44	Random	N/A	N/A	-	1.60	0.05	Random
42	0.73	0.78	Random	0.72	0.77	Random	1.87	0.03	Aggregated
59	0.59	0.99	Uniform	1.41	0.10	Random	2.59	0.01	Aggregated

* N/A indicates that insect counts were insufficient to conduct aggregation analysis.

## Data Availability

All data included in this study are available upon request by contact with the corresponding author.

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
