# Peer review of "Impact of *Zinnia elegans* Cultivation on the Control Efficacy and Distribution of *Aphidius colemani* Viereck (Hymenoptera: Braconidae) against *Aphis gossypii* Glover (Hemiptera: Aphididae) in Cucumber Greenhouses"

_insects, 2024, doi:10.3390/insects15100807_

Round 1

Reviewer 1 Report

Comments and Suggestions for Authors

Dear authors, here are my comments:

1. Introduction:

-You should add a hypothesis at the end of the introduction about the role of Zinnia elegans on cucumber biological control.

2. Materials and Methods:

-Line 91: You have two seasons and two years, I do not understand your design. Are you comparing them?

-Line 96: it's not clear the number of greenhouses per treatment. I understand that one for each treatment. You should explain it better.

-Line 117: To analyze data over time, an ANOVA on each time is not the best option. You should use time series analysis in mixed effect models, since there is an accumulative effect and also random factors like time and greenhouse.

-Line 123: I would use all the season and years, and time as a random factor.

3. Results:

-Figure 3: One of the reason that your statistical analysis is not correct is shown here. Hw do you explain that at 42 DACT there are differences and at 59 there are not?

-Figure 5: Change spring season for fall season.

-Table 1: I strongly recommend analyzing this data with a time series analysis in mixed effect models.

4. Discussion:

-Line 303: The trend is also present in 2017 and in falls season, you may add this sentence.

Author Response

Comments 1. (Introduction) You should add a hypothesis at the end of the introduction about the role of Zinnia elegans on cucumber biological control.

Thank you for your suggestion. We added the related contents in the manuscript. The change can be found in the end of Introduction (line 77).

Comments 2. (Material and Methods, Line 91) You have two seasons and two years, I do not understand your design. Are you comparing them?

Comments 3. (Material and Methods, Line 96) It's not clear the number of greenhouses per treatment. I understand that one for each treatment. You should explain it better.

Thank you for your valuable feedback on our study. We appreciate your insight regarding the limitations of our experimental design, particularly, concerning the allocation of treatments to individual greenhouses and the potential confounding effects of greenhouse specific variations.

Due to constraints in conducting the study directly on farms and the limited available of greenhouses, we performed the cultivation and experiments and simultaneously within the greenhouse available. As a result, each treatment was assigned to a separate greenhouse, and sampling was conducted on cucumber plants within the greenhouses. While we acknowledge that the effects of the greenhouse and treatment may be confounded, we addressed this by conducting four trials with random allocation of greenhouses in each trial. We believe that the random assignment of greenhouses across four trials provided sufficient consistency in results, allowing us to reasonable compare the differences between treatments. The repeated randomization of greenhouse assignments across trials helped mitigate the confounding effects to some extent, thereby supporting the validity of our findings regarding treatment differences.

We have added additional explanations of the experimental setup in the Materials and Methods section, detailing the greenhouse assignments in line 118.

Comments 4. (Material and Methods, Line 117) To analyze data over time, an ANOVA on each time is not the best option. You should use time series analysis in mixed effect models, since there is an accumulative effect and also random factors like time and greenhouse.

Comments 5. (Material and Methods, Line 123) I would use all the season and years, and time as a random factor.

All authors in this manuscript discussed your valuable suggestion to use time series analysis instead of ANOVA analysis. We think that the ANOVA analysis might be better than the time series analysis in this study. There are multiple reasons. First of all, the effects should be accumulated. The mummy is not just dead aphid, but a growing wasp. As time goes, this wasp becomes adults and parasites healthy aphids. Second, the effects should be consistent regardless of times and greenhouses even though the required times (days) to be effective should be different according to the seasons and greenhouses.

Comments 6. (Figure 3) One of the reason that your statistical analysis is not correct is shown here. How do you explain that at 42 DACT there are differences and at 59 there are not?

→ The main reasons of non-significance at 59 DACT were high variations of parasitism rate (%) among treatments and rapidly increased parasitism rate between 42 days to 59 days. During the rapid increase, the searching time (to find healthy aphid) becomes more important rather than the handling time (to hand its preys). This mean the density of healthy aphids could be main factor to affect parasitism rate. Moreover, the trend at 42 DACT was not changed compared to one at 59 DACT. 

Comments 7. (Figure 5) Change spring season for fall season.

→Thank you for pointing out the error in the table title. We have corrected spring to fall.

Comments 8. (Table 1) I strongly recommend analyzing this data with a time series analysis in mixed effect models.

→ All authors in this manuscript discussed your valuable suggestion to use time series analysis instead of ANOVA analysis. We think that the ANOVA analysis might be better than the time series analysis in this study. There are multiple reasons. First of all, the effects should be accumulated. The mummy is not just dead aphid, but a growing wasp. As time goes, this wasp becomes adults and parasites healthy aphids. Second, the effects should be consistent regardless of times and greenhouses even though the required times (days) to be effective should be different according to the seasons and greenhouses.

Comments 9 (Discussion) The trend is also present in 2017 fall season, you may add this sentence

→ Thank you for your suggestion. In the response to the comment suggesting the addition of a sentence related to the 2017 fall season, we have revised the sentence as follows: 

“This trend was also observed in other three trials where the density of A. gossypii in the parasitoid-zinnia plot was significantly lower compared to other treatment plots.”

Reviewer 2 Report

Comments and Suggestions for Authors

In this work, the researchers evaluated the effect of a companion plant (Zinnia elegans) on the efficiency of the biological controller Aphidius colemanii on Aphis gosypii in cucumber crops. Zinnia provides resources that favor the establishment of the parasitoid, and the authors manage to demonstrate the much greater effectiveness of the parasitoid by using Zinnia as a companion plant. The work is simple, and concrete and has a clear practical application to improve the use of biological control.

Authors must improve some aspects before publishing the work:

Lines 40-80: Is this the introduction?

Line 40: Change glover by Glover

Lines 117-119: Which experimental design was used? What was the experimental unit? Do the sampled plants come from a unique area with each treatment? If so, a pseudo-replication effect may occur.

Line 122: This transformation is used mainly to stabilize the variance, which is not constant for proportions: Often, when the data is proportions, the variance is smallest near 0 and 1 and largest near 0.5

Line 275: What is Pa? Explain the abbreviation when first used

Line 339: prepositions? what do you mean?

Author Response

Comments 1. (Line 40-80) Is this the introduction?

→ Thank you for pointing this out. We insert the introduction section title.

Comments 2. (Line 40) Change glover by Glover

→ Thank you for pointing this out. We have changed glover to Glover.

Comments 3. (Lines 117-119: Which experimental design was used? What was the experimental unit? Do the sampled plants come from a unique area with each treatment? If so, a pseudo-replication effect may occur.

→ Thank you for your valuable feedback on our study. We appreciate your insight regarding the limitations of our experimental design, particularly, concerning the allocation of treatments to individual greenhouses and the potential confounding effects of greenhouse specific variations.

Due to constraints in conducting the study directly on farms and the limited available of greenhouses, we performed the cultivation and experiments and simultaneously within the greenhouse available. As a result, each treatment was assigned to a separate greenhouse, and sampling was conducted on cucumber plants within the greenhouses. While we acknowledge that the effects of the greenhouse and treatment may be confounded, we addressed this by conducting four trials with random allocation of greenhouses in each trial. We believe that the random assignment of greenhouses across four trials provided sufficient consistency in results, allowing us to reasonable compare the differences between treatments. The repeated randomization of greenhouse assignments across trials helped mitigate the confounding effects to some extent, thereby supporting the validity of our findings regarding treatment differences.

We have added additional explanations of the experimental setup in the Materials and Methods section, detailing the greenhouse assignments. Furthermore, we have expanded section to thoroughly address the limitations of our analysis.

Comments 4. (Lines 122): This transformation is used mainly to stabilize the variance, which is not constant for proportions: Often, when the data is proportions, the variance is smallest near 0 and 1 and largest near 0.5.

→ Thank you for your comment regarding the data transformation. Our data did not follow the normality assumption of the ANOVA test. After the arcsine transformation, our data satisfied the normality assumption. That’s why we used the arcsine data transformation.  

Comments 5. (Line 275) What is Pa? Explain the abbreviation when first used.

→ Thank you for pointing out the omission. I have added an explanation for Pa in the Materials and Methods section. The sentence would be:

“The associated probability (Pa) of this aggregation index was calculated using randomization tests at P = 0.05.”

Comments 6. (Line 339) preposition? What do you mean?

→ There was an error. It’s “preposition”, not “proposition”. We corrected it.

Reviewer 3 Report

Comments and Suggestions for Authors

The manuscript "Impact of Zinnia elegans cultivation on the control efficacy and distribution of Aphidius colemani Viereck (Hymenoptera: Braconidae) against Aphis gossypii Glover (Hemiptera: Aphididae) in cucumber greenhouses" is a good work on the biological control of A. gossypii. The manuscript is written well but needs much improvement in the results section and the discussion. Various results are misinterpreted and should be rewritten and concise. The manuscript can be considered with significant revision. My comments and suggestions are given in the attached pdf.

Comments on the Quality of English Language

Moderate improvement in English needs by a native speaker.

Author Response

Comments 1. (Line 58) Provide all references here

→ Thank you for pointing this out. We have combined references in the end of the sentences.

Comments 2. (Line 86, 97) Provide specification of plastic greenhouse

→ Thank you for your comment. The specification of the plastic greenhouse are already detailed in the Materials and Methods section, as follows:

“Each greenhouse unit, covering an area of 260 m2, was internally partitioned into four rows, each 0.8 m wide with a 0.4 m gap between them. Mesh screens were fitted on both sides of the greenhouse and at the front and rear entrances.”

Comments 3. (Line 109-116) Which experimental design was used?

→ We used “completely randomized experimental design”. We added few sentences according to the comments of other reviewers. We hope this is helpful to clarify the experimental design of our study.

Comments 4. (Line 124) Why data of 2016 was used only, why not 2017?

→ The results were similar, but it required large space. That’s why we did not mention the results of 2017. We think this information is helpful, but not required.

Comments 5. (Line 143, 162, 252) This result show significant difference as P<0.05, Check it

→ We checked and corrected them.

Comments 6. (Line 167) Rephrase sentence: In the parasitoid plot, A. gossypii remained low until 43 DACT Subsequently increasing to 493 individuals per leaf by 63 DACT.  

→ Thank you for your suggestion to rephrase the sentence. We have revised it for clarity and precision as follows:

“In the parasitoid plant, A. gossypii remained low until 43 DACT but subsequently increased, reaching 493 individuals per leaf by 36 DACT.” 

Comments 7. (Line 191) Delete sentence: This was followed by the parasitoid, and the control.

→ Thank you for your suggestion. We have deleted the sentence as recommended to make the manuscript more concise and improve its clarity.

Comments 8. (Line 295) Cite Table 1 in the text.

→ Table 1 has been cited in the manuscript on line 227.

Comments 9. (Line 322) Provide authority name of Aphidis rhopalosiphi and Ciaeretiella rapae.

→ The authority name of the insects have been added to the manuscript as requested.

Round 2

Reviewer 1 Report

Comments and Suggestions for Authors

Dear authors,

I agree with the modification of introduction adding a hypothesis. However, I disagree with your explanation about the statistical analysis. In your reasoning, you accept that your data is changing over time. But you said that is better using an ANOVA for each sampling. This is a contradiction, since time series analysis is used for this kind of data.

Author Response

Comment.  I disagree with your explanation about the statistical analysis. In your reasoning, you accept that your data is changing over time. But you said that is better using an ANOVA for each sampling. This is a contradiction, since time series analysis is used for this kind of data. 

→ We respect your suggestion. According to your suggestion, we applied our data to one of time series analysis methods. As your expectation, the results of statistical analyses were more consistent and clearer than the ones in previous method. Sincerely, thank you for your suggestion. According to this changes, ‘material & method’, ‘results’, and ‘figures’ in the manuscript were changed and marked.